behaviour/biochemistry/evolution

(*Z*)-11-eicosen-1-ol, *Apis mellifera*, honeybees, colony collapse, hive health, *Nosema ceranae*

**Author for correspondence:**
Christopher Mayack
e-mail: cmayack@sabanciuniv.edu

# Increased alarm pheromone component is associated with *Nosema ceranae* infected honeybee colonies

Christopher Mayack[1,2], Robert L. Broadrup[3,4], Sassicaia J. Schick[2], Elizabeth J. Eppley[2], Zaeema Khan[1] and Anthony Macherone[5,6]

[1]Molecular Biology, Genetics and Bioengineering Program, Faculty of Engineering and Natural Sciences, Sabanci University, Tuzla, Istanbul, Turkey
[2]Biology Department, Swarthmore College, 500 College Avenue, Swarthmore, PA, USA
[3]Department of Chemistry, Haverford College, Haverford, PA, USA
[4]Department of Chemistry, Lafayette College, Easton, PA 18042, USA
[5]Life Science and Chemical Analysis Group, Agilent Technologies, Santa Clara, CA 95051, USA
[6]Department of Biological Chemistry, The Johns Hopkins University School of Medicine, Baltimore, MD, USA

CM, 0000-0003-0213-2149; ZK, 0000-0002-7656-5265; AM, 0000-0002-0440-8367

Use of chemicals, such as alarm pheromones, for rapid communication with conspecifics is widespread throughout evolutionary history. Such chemicals are particularly important for social insects, such as the honeybee (*Apis mellifera*), because they are used for collective decision-making, coordinating activities and self-organization of the group. What is less understood is how these pheromones change due to an infection and what the implications might be for social communication. We used semiquantitative polymerase chain reaction (sqPCR) to screen for a common microsporidian gut parasite, *Nosema ceranae*, for 30 hives, across 10 different locations. We then used high-resolution accurate mass gas chromatography–quadrupole time of flight mass spectrometry to generate an exposome profile for each hive. Of the 2352 chemical features identified, chemicals associated with infection were filtered for cosanes or cosenes. A significant association was found between *N. ceranae* and the presence of (*Z*)-11-eicosen-1-ol, a known alarm pheromone component. The increase in (*Z*)-11-eicosen-1-ol could be the recognition mechanism for healthy individuals to care for, kill, or quarantine infected nestmates. *Nosema ceranae* has contributed to the global decline in bee health. Therefore, altered alarm pheromones might play a role in disrupting social harmony and have potential impacts on colony health.

# 1. Background

The use of pheromones and cuticular hydrocarbons for communication is common among insects. Consequently, their olfactory systems are well adapted to detect these chemical compounds [1]. The function of pheromones for communication is highly conserved in insects and probably traces back at least 420 million years, pre-dating the divergence of Crustacea and Hexapoda [2]. Social insects behave as a highly organized colony due to the remarkable chemical communication within the hive. Pheromones and cuticular hydrocarbons like pentacosane (*n*-C25) to tritriacontane (*n*-C33) are instrumental in coordination of hive activities among nestmates and nestmate recognition, respectively, for optimal functioning of the colony [3–6].

Honeybee alarm pheromones are released after a bee is crushed or stings an intruder and triggers other bees to engage the trespasser to protect the hive [7]. Guard and forager bees are known to be particularly sensitive to alarm pheromone components, such as 2-heptanone and (*Z*)-11-eicosen-1-ol [8,9]. However, the impact on social behaviour, whether to attract or repel a nestmate, depends on the context of the presentation of the component [10–12]. If a bee is under stress, then the pheromone component can act as a repelling signal to other nestmates, similar to when nestmates are crushed or killed [10]. Alternatively, foraging bees may use the same component to attract other nestmates to profitable foraging sites by marking these with the same pheromone component [11].

What is less clear is how the alarm pheromone profile produced might be altered during infection and what consequences this might have for social communication. To date, there are sparse examples in the literature that have demonstrated that infections can alter alarm pheromone production. One example has been demonstrated with the kissing bug, *Triatoma infestans*, a fungal pathogen that stimulates increased propionic acid biosynthesis which is a main component of the bug's alarm pheromone. The study suggests that this chemical cue may aid in social distancing and reduce transmission of the pathogen between conspecifics [13].

A number of pheromone changes have been observed for honeybee workers and queens due to *Nosema ceranae* fungal infection as well [14,15]. *Nosema ceranae* is a microsporidian fungal gut pathogen, and it has played a significant role in the most recent decline of bee health. *Nosema ceranae* is widely found in *Apis mellifera*, the European honeybee, around the world and appears to be displacing the once prevalent *Nosema apis* [16]. While *N. ceranae* typically does not cause the demise of a hive on its own and is not very virulent compared with other pathogens, it does cause a number of physiological and behavioural alterations in hunger [17], immune functioning [18], metabolic pathways [19], hormone production [20,21] and neurotransmitter composition [22]. These changes associated with accelerated age polyethism and shortened lifespan [23], can weaken entire colonies and eventually cause them to collapse [24]. In this study, we employed high-resolution accurate mass (HRAM) gas chromatography–quadrupole time of flight (GC-QTOF) analysis to characterize volatile chemicals in honeybee exposomes and semiquantitative polymerase chain reaction (sqPCR) to determine the *N. ceranae* infection status of the colony. This approach was specifically used to capture and measure known as well as unknown pheromones that have yet to be characterized. We filtered this data so that we could identify putative associations between alarm pheromones and infection.

# 2. Methods

## 2.1. Sample collection

Roughly 100 forager bees were collected from 30 different hives, across 10 different locations in the greater Philadelphia, PA, USA, region. Bees were immediately stored on dry ice in 50 ml Falcon tubes. These samples were thawed and homogenized in 6 ml of DNase- and RNase-free water using a sterile 50 ml tissue grinder (Fisher Scientific). The homogenate was divided into two groups: one for sqPCR, and the other for GC-QTOF analysis. All samples were stored at −80°C until further analysis.

## 2.2. Semiquantitative polymerase chain reaction screening for *Nosema* spp.

DNA was extracted from honeybees per the Honey Bee Research Centre (HBRC) method detailed by Hamiduzzaman *et al.* [25]. A total of 300 µl of 0.03 M hexadecyltrimethyl ammonium bromide, 0.05 M tris-hydroxymethyl aminomethane, 0.01 M ethylenediamine tetra-acetic acid and 1.1 M NaCl, in d $H_2O$ (extraction buffer) were added to 150 µl of bee homogenate, and the resulting mixture was then mixed

and macerated using a sterile pestle. Then a phenol-chloroform extraction was performed (for details, please see electronic supplementary material). The extracted DNA was quantified using a Nanodrop 2000 UV–Vis Spectrophotometer (Thermo Scientific) and further diluted to a working concentration of 5 ng μl$^{-1}$ for subsequent PCRs.

A total of two duplex reactions were performed on two different subsamples, one screening for *N. apis* and the other for *N. ceranae*. The final PCR volume was 15 μl. The *Nosema*-specific primers used can be found in Hamiduzzaman *et al.* [25]. The duplex PCR reaction consisted of: 1 μl of a 10 mM solution of each of the four primers (4 μl total volume), 1.5 μl of 10× PCR buffer, 0.5 μl of 10 mM dNTPs, 0.2 μl of 25 mM MgCl$_2$ and 0.2 μl of 5 U μl$^{-1}$ Taq DNA polymerase (New England BioLabs, Ipswich, MA) with an additional 2 μl of template DNA and 6.6 μl sterile Millipore water. The reference gene used was RpS5 [25]. The PCR thermocycler programme was: 94°C for 2.5 min; 10 cycles of 15 s at 94°C, 30 s at 61.8°C and 45 s at 72°C, and 20 cycles of 15 s at 94°C, 30 s at 61.8°C and 50 s at 72°C, an extension step at 72°C for 7 min, and a final hold step of 4°C. Each PCR assay contained a negative and positive control for each *Nosema* species. PCR products were run on a 3% agarose gel at 100 V for 3.5 h (New England BioLabs, Ipswich, MA, USA) and semiquantified using ImageJ software [26]. Each sample was run in triplicate, and the relative amount of DNA was averaged across the runs.

## 2.3. GC-QTOF analysis

A QuEChERS extraction method was followed for the GC-QTOF sample preparation (for details, see the electronic supplementary material) [27]. For non-targeted profiling of honeybee extracts, an Agilent 7890B/7200B GC-QTOF system was used. A 0.2 μl pulsed split-less injection was made using a split/splitless inlet at 250°C. A 40 m × 0.25 mm × 0.25 μm DB5-MS DuraGuard column (J&W 122–5532G) and was installed and operated at a 1.2 ml min$^{-1}$ flow rate with helium carrier gas. The oven was held at 80°C for 1 min, then ramped 10°C min$^{-1}$ to 310°C, followed by a 6 min hold time. The temperature of the transfer line was 300°C. The mass spectrometer was operated in electron ionization, high resolution mode. The temperatures of the source and quadrupole (RF only) were 275°C and 150°C, respectively. We collected HRAM spectral data at 5 Hz over a mass range of 50–800 Da. Immediately prior to each sample injection, an automated intra-sequence mass calibration was performed. We used the average of duplicate runs for data analysis.

## 2.4. Chromatographic deconvolution and chemical entity annotation

We used the MassHunter suite software to analyse the raw data acquired by the GC-QTOF system. Unknowns Analysis B.08.00 was used to perform chromatographic deconvolution. The cut-off for including chemical features was a signal-to-noise ratio of greater than or equal to 3 : 1, and an accurate mass assignment was needed for the base ion that was associated with a retention time for a particular chromatographic peak.

We conducted spectral library searches and compound annotation using the RTL Pesticide, the Fiehn Metabolomics (Agilent Technologies, Santa Clara, CA, USA), and the NIST-11 Mass Spectral Libraries (the National Institute of Standards and Technology, NIST Standard Reference Database 1A v. 11). We were unable to identify 629 chemical features based on the minimal feature parameters defined above, so a composite mass spectrum was used instead for a covariate statistical analysis.

## 2.5. Statistical testing and covariate analysis

To identify chemical features associated with the *N. ceranae*-infected bee hives, we used Agilent MassProfiler Professional bioinformatics software. We first aligned the retention times of each compound and based on the median abundance of all chemical entities, we then established a baseline. Lastly, we filtered out features that had a relative ion abundance fold-change that was less than at least three times relative to the median ion abundance calculated across all samples. We conducted an unpaired, unequal variance *T*-test to compare the relative abundance of each chemical found in uninfected and infected hives. We corrected for multiple testing using the Benjamini–Hochberg false discovery rate (FDR) method and set an alpha level of 0.05.

# 3. Results and discussion

The mean (± s.d.) *N. ceranae* load was 547 045 spores bee$^{-1}$ ± 1 324 278 spores bee$^{-1}$. There was a total of 18 hives with a detectable *N. ceranae* load, and these were considered infected hives. The remaining 12

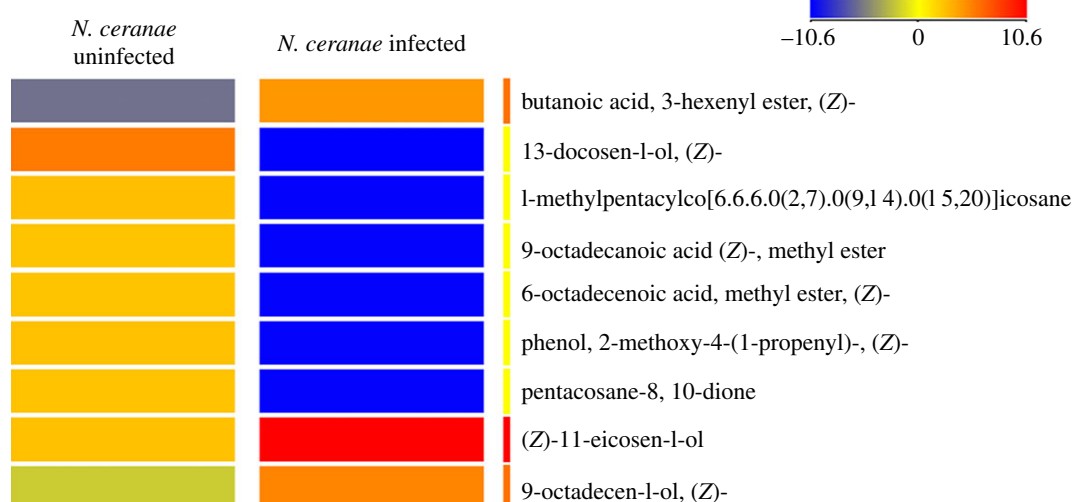

**Figure 1.** A heatmap comparing the relative abundance of the compounds filtered in search of the annotated honeybee pheromones obtained from the *N. ceranae* infected and uninfected hives. A total of nine possible pheromones, out of the 13 annotated by GC–MS analysis were taken to generate the heatmap using the Agilent Mass Profiler Professional software. The colours next to the labels indicate the fold changes across both the infected and uninfected hives relative to the median of all of the compounds detected. Red indicates a relatively higher abundance, blue indicates a relatively lower abundance and yellow indicates a neutral change in relative abundance.

hives did not contain any *N. ceranae* load and were therefore considered uninfected hives. Despite screening for *N. apis,* none was detected in any of the 30 hives.

We identified a total of 2352 chemical features with a signal to noise ratio of greater than or equal to 3 : 1 after chromatographic deconvolution. Of these chemical features, 1723 (73%) were annotated (retention time, ion abundance, *m/z*, chemical name, CAS number) from the commercial databases. We identified the top 13 annotated compounds based on the highest overall fold change abundances that were statistically associated with *N. ceranae* infection. While most of the compounds were found to have a relatively lower abundance in infected hives, butanoic acid, 3-hexenyl ester, (Z)-11-eicosen-1-ol and 9-Octadecen-1-ol, (Z)- had relatively higher abundance (the top nine are shown in figure 1). These data revealed a significantly higher relative abundance of (Z)-11-eicosan-1-ol in *N. ceranae* infected versus uninfected hives ($T = -1.76$, d.f. $= 29$, $p = 0.045$) (figure 2). (Z)-11-eicosan-1-ol was manually identified based on previous literature because spectral ions and ion ratios did not exist in the NIST database [28].

Compared with the other identified compounds and their possible role as pheromones or cuticular hydrocarbons, only (Z)-11-eicosen-1-ol (Z-11-ol), was found to have a significant association with *N. ceranae* infection at the colony level. As this is one of the primary components of the alarm pheromone system in the honeybee, its increased presence in the hive may result in behavioural alterations [12]. However, it is also possible that, as with aphids infected with a fungal pathogen [29], infected honeybees are less sensitive to alarm pheromone and, consequently, conspecifics produce more of the compound to compensate, resulting in no behavioural changes. In addition, we cannot rule out that the increased alarm pheromone from the infected bee colonies observed may be a by-product of increased ageing in the honeybee as infected bees age faster and alarm pheromone production increases with age [23].

(Z)-11-eicosen-1-ol is a 20 carbon-chain primary alcohol with a single double bond at $C_{11}$. This alarm pheromone component is known to be relatively less volatile in comparison with other components and has also been identified as a cuticular hydrocarbon of honeybees. Our findings are in agreement with previous work that shows *N. ceranae*-infected bees can produce different chemical compounds used for communication in comparison with uninfected bees. In previous studies, the altered hydrocarbon profiles of infected bees were not associated with any significant behavioural changes towards infected individuals. However, no differences were found in regard to (Z)-11-eicosen-1-ol production like in our study [30,31]. Higher amounts of this alarm pheromone component produced at the colony

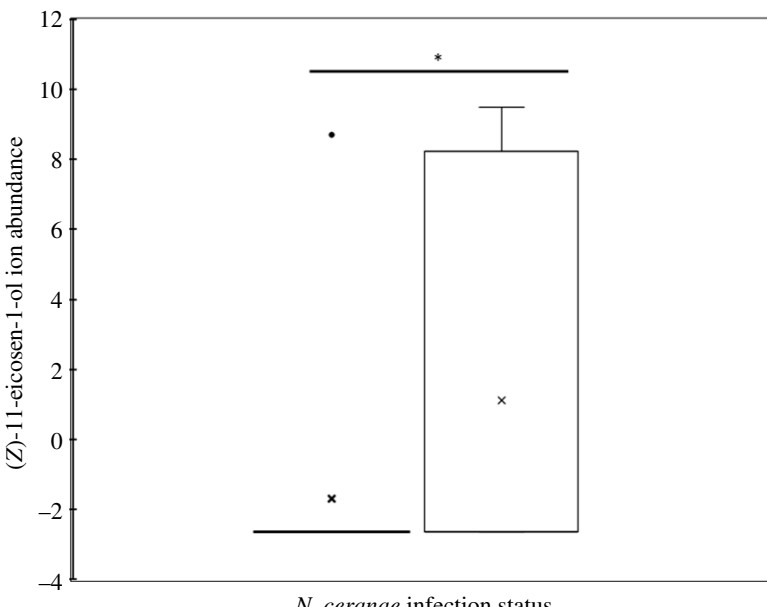

**Figure 2.** Box and whisker plot of the (Z)-11-eicosen-1-ol log2-transformed ion abundance comparing *Nosema ceranae* infected and uninfected hives. The 'x' indicates the median of the log transformed data, while the black dot indicates an outlier for one of the hives. The box represents the interquartile range (IQR), and the error bar represents the maximum value. The asterisk (*) indicates a significant difference between infected and uninfected hives at the alpha = 0.05 level.

level could have far-ranging effects, as it even mediates communication in plant–pollinator–predator interactions (see [7]). More broadly, other alarm pheromone components have been shown to decrease foraging and waggle dancing in honeybees [32].

## 4. Conclusion

Further investigation is needed to confirm the possible behavioural effects coming from the association between an *N. ceranae* infection and an increased (Z)-11-eicosen-1-ol alarm pheromone component. Healthy individuals have recently been demonstrated to increase their interaction and sometimes kill an *N. ceranae* infected individual. However, according to this previous study, if the hive is infected as a whole, infected bees may maintain their distance from one another instead [33]. Both aggressive and repelling behaviours have been documented with the (Z)-11-eicosen-1-ol alarm pheromone, as well as with viral infections and general immune responses in bees, depending on the context [34]. Yet, the driving mechanism of these care, kill, or quarantine alternatives for a social behavioural response to an *N. ceranae* infection remains unknown, but it is plausible that (Z)-11-eicosen-1-ol could be one such driver. Furthermore, social immunity with an alarm system in response to a fungal infection has been documented before in termites [35], therefore such an alarm system in honeybees is also plausible, but further investigation is needed to confirm this.

Data accessibility. All raw data are provided as an electronic supplementary material file under the 'Raw Data' file name. The data are provided in the electronic supplementary material [36].

Authors' contributions. Funding to carry out and the original conceptualization of the study was obtained by C.M., R.L.B. and A.M. C.M., A.M. and S.J.S. conducted the data analysis. C.M., S.J.S., E.J.E., R.L.B. and A.M. were responsible for carrying out the study and collecting the data. C.M. and Z.K. were primarily responsible for writing the manuscript. All authors contributed to reviewing and editing the manuscript.

Competing interests. We declare we have no competing interests.

Funding. This work was supported by an Agilent Applications and Core Technology University Research grant no. (3937) to C.M., R.L.B. and A.M.

Acknowledgements. We thank Chloe Wang, Malia Wenny, Alexis Schafsnitz, Naomi Chaqueco and Katiana Rufino of Haverford College and Rebecca Zhou of Swarthmore College for their assistance in the field and the laboratory. We also thank Profs. Christina Grozinger and Robert J. Paxton for providing positive controls of *N. ceranae* and *N. apis*, respectively.

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
