## [Peer Review File · Royal Society Open Science]

Review History

RSOS-210194.R0 (Original submission)

Review form: Reviewer 1

Is the manuscript scientifically sound in its present form?

Yes

Are the interpretations and conclusions justified by the results?

Yes

Is the language acceptable?

Yes

Do you have any ethical concerns with this paper?

No

Have you any concerns about statistical analyses in this paper?

No

Recommendation?

Accept with minor revision (please list in comments)

Comments to the Author(s)

The manuscript is now improved but they still need to be cautious with the interpretation of results in the abstract. In line 46, they mention this increase in alarm pheromone production might drive the killing of infected nestmates. There is no evidence for that. In addition, in the discussion they mention different outcomes for this pheromone change, such as more care, kill or quarantine. However, in the abstract they only cite the killing hypothesis. They should therefore either remove it from the abstract or fairly report all possible outcomes.

The following reference, where a change in CHCs was found upon *Nosema* infection, is missing. No change in behavioural interactions was found.

McDonnell, C.M., Alaux, C., Parrinello, H. et al. Ecto- and endoparasite induce similar chemical and brain neurogenomic responses in the honey bee (*Apis mellifera*). *BMC Ecol* 13, 25 (2013)

Line 174 : it is still not clear what is the amount of spores ; 3.99 millions? If yes, millions should be added. In addition, what do the numbers in parenthesis mean? 3.99 (547,045 spores/bee)? I am sorry but I am confused by how the number of spores are reported.

Line 219: "sometimes kill an *N. ceranae* infected individual", is there evidence for that? A reference?

Decision letter (RSOS-210194.R0)

Dear Dr Mayack

On behalf of the Editors, we are pleased to inform you that your Manuscript RSOS-210194 "Increased alarm pheromone component is associated with *Nosema ceranae* infected honey bee colonies" has been accepted for publication in Royal Society Open Science subject to minor revision in accordance with the referees' reports. Please find the referees' comments along with any feedback from the Editors below my signature.

Please submit your revised manuscript and required files (see below) no later than 7 days from today's (ie 07-Apr-2021) date. Note: the ScholarOne system will 'lock' if submission of the revision is attempted 7 or more days after the deadline. If you do not think you will be able to meet this deadline please contact the editorial office immediately.

Please note article processing charges apply to papers accepted for publication in Royal Society Open Science (<https://royalsocietypublishing.org/rsos/charges>). Charges will also apply to papers transferred to the journal from other Royal Society Publishing journals, as well as papers

submitted as part of our collaboration with the Royal Society of Chemistry (<https://royalsocietypublishing.org/rsos/chemistry>). Fee waivers are available but must be requested when you submit your revision (<https://royalsocietypublishing.org/rsos/waivers>).

on behalf of Prof Kevin Padian (Subject Editor)
openscience@royalsociety.org

Editor comments:

Thanks for your efforts in revision. We are happy to accept your manuscript but we would like you to address specifically the comments of the reviewer in your final version before we can process it. Best wishes.

Reviewer comments to Author:

Reviewer: 1

Comments to the Author(s)

The manuscript is now improved but they still need to be cautious with the interpretation of results in the abstract. In line 46, they mention this increase in alarm pheromone production might drive the killing of infected nestmates. There is no evidence for that. In addition, in the discussion they mention different outcomes for this pheromone change, such as more care, kill or quarantine. However, in the abstract they only cite the killing hypothesis. They should therefore either remove it from the abstract or fairly report all possible outcomes.

The following reference, where a change in CHCs was found upon *Nosema* infection, is missing. No change in behavioural interactions was found.

McDonnell, C.M., Alaux, C., Parrinello, H. et al. Ecto- and endoparasite induce similar chemical and brain neurogenomic responses in the honey bee (*Apis mellifera*). *BMC Ecol* 13, 25 (2013)

Line 174 : it is still not clear what is the amount of spores ; 3.99 millions? If yes, millions should be added. In addition, what do the numbers in parenthesis mean? 3.99 (547,045 spores/bee)? I am sorry but I am confused by how the number of spores are reported.

Line 219: "sometimes kill an *N. ceranae* infected individual", is there evidence for that? A reference?

===PREPARING YOUR MANUSCRIPT===

===PREPARING YOUR REVISION IN SCHOLARONE===

- If you are providing image files for potential cover images, please upload these at this step, and inform the editorial office you have done so. You must hold the copyright to any image provided.
- A copy of your point-by-point response to referees and Editors. This will expedite the preparation of your proof.

- Ensure that your data access statement meets the requirements at <https://royalsociety.org/journals/authors/author-guidelines/#data>. You should ensure that you cite the dataset in your reference list. If you have deposited data etc in the Dryad repository, please only include the 'For publication' link at this stage. You should remove the 'For review' link.
- If you are requesting an article processing charge waiver, you must select the relevant waiver option (if requesting a discretionary waiver, the form should have been uploaded at Step 3 'File upload' above).
- If you have uploaded ESM files, please ensure you follow the guidance at <https://royalsociety.org/journals/authors/author-guidelines/#supplementary-material> to include a suitable title and informative caption. An example of appropriate titling and captioning may be found at https://figshare.com/articles/Table_S2_from_Is_there_a_trade-off_between_peak_performance_and_performance_breadth_across_temperatures_for_aerobic_scope_in_teleost_fishes_/3843624.

Author's Response to Decision Letter for (RSOS-210194.R0)

See Appendix A.

Decision letter (RSOS-210194.R1)

Dear Dr Mayack,

I am pleased to inform you that your manuscript entitled "Increased alarm pheromone component is associated with *Nosema ceranae* infected honey bee colonies" is now accepted for publication in Royal Society Open Science.

on behalf of Professor Kevin Padian (Subject Editor)
openscience@royalsociety.org

Appendix A

Response to Reviewer Comments

Dear Prof Kevin Padian and reviewer,

We would like to thank you for the detailed comments and feedback from the Royal Society Open journal submission. We realize that the reviewer process is voluntary and appreciate the time you have taken to review this manuscript.

We have made alterations to the abstract, results, discussion and added reference as suggested. With all of these comments addressed we now feel that our manuscript is improved. Below you will find a point by point response for the remaining issues that have been raised in italicized text. We look forward to hearing from you once again and are pleased to hear that the paper has been accepted for publication.

Sincerely,

Christopher Mayack

Reviewer comments to Author:

Reviewer: 1

Comments to the Author(s)

The manuscript is now improved but they still need to be cautious with the interpretation of results in the abstract. In line 46, they mention this increase in alarm pheromone production might drive the killing of infected nestmates. There is no evidence for that. In addition, in the discussion they mention different outcomes for this pheromone change, such as more care, kill or quarantine. However, in the abstract they only cite the killing hypothesis. They should therefore either remove it from the abstract or fairly report all possible outcomes.

We agree with the reviewer that to be fair all possible outcomes should be stated in the abstract. We have now included the other possible outcomes in the abstract.

The following reference, where a change in CHCs was found upon Nosema infection, is missing. No change in behavioural interactions was found.

McDonnell, C.M., Alaux, C., Parrinello, H. et al. Ecto- and endoparasite induce similar chemical and brain neurogenomic responses in the honey bee (*Apis mellifera*). BMC Ecol 13, 25 (2013)

Thanks for pointing this out, we have now added this reference.

Line 174 : it is still not clear what is the amount of spores ; 3.99 millions? If yes, millions should be added. In addition, what do the numbers in parenthesis mean? 3.99 (547,045 spores/bee)? I am sorry but I am confused by how the number of spores are reported.

Now we understand where the confusion is coming from and this is because we performed a semi-quantitative analysis and the number refers to a relative quantification of the amount of spores with a unitless scale. At the same time, we report the spore numbers based on a standard curve generated from a previous study. To avoid confusion, we have removed the less informative semi-quantitative number.

Line 219: “sometimes kill an *N. ceranae* infected individual”, is there evidence for that? A reference?

Yes, there is evidence in the article that is cited in the manuscript which can also be found below...

[33] Biganski, S., Kurze, C., Müller, M.Y. & Moritz, R.F.A. 2018 Social response of healthy honeybees towards *Nosema ceranae*-infected workers: care or kill? *Apidologie* **49**, 325-334. (doi:10.1007/s13592-017-0557-8).